# Preparations from Various Organs of Sea Buckthorn (*Elaeagnus rhamnoides* (L.) A. Nelson) as Important Regulators of Hemostasis and Their Role in the Treatment and Prevention of Cardiovascular Diseases

**DOI:** 10.3390/nu14050991

**Published:** 2022-02-26

**Authors:** Beata Olas, Bartosz Skalski

**Affiliations:** Department of General Biochemistry, Faculty of Biology and Environmental Protection, University of Łódź, Pomorska 141/143, 90-236 Łódź, Poland; bartosz.skalski@biol.uni.lodz.pl

**Keywords:** food product, hemostasis, platelets, safety, sea buckthorn

## Abstract

Numerous studies on the chemical composition of various organs of sea buckthorn (*Elaeagnus rhamnoides* (L.) A. Nelson) have found the plant to be a rich source of vitamins, phenolic compounds, amino acids, fatty acids, and micro- and macro-elements. Furthermore, other studies on preparations from various organs have found them to have significant anti-cancer, anti-ulcer, and hepatoprotective properties, as well as various antibacterial and antiviral activities. This paper reviews the current literature concerning the effect of different sea buckthorn preparations, i.e., extracts and fractions with various chemical contents, on hemostasis, and their positive role in the treatment and prevention of cardiovascular diseases. It also sheds new light on the mechanisms involved in their action on hemostasis both in vivo and in vitro. For these studies, biological materials, including blood platelets, plasma, and blood, were isolated from healthy subjects and those with cardiovascular risk factors. In addition, it describes the cardioprotective potential of commercial products from different organs of sea buckthorn.

## 1. Introduction

Sea buckthorn (*Elaeagnus rhamnoides* (L.) A. Nelson) can grow to the size of a small tree, and it belongs to the olive family (*Elaeagnaceae*). It naturally occurs in Russia, China, the Caucasus, and Siberia, mainly along the seacoasts [1,2,3].

Numerous in vitro and in vivo studies on the biological activity of preparations obtained from various sea buckthorn organs indicate that they may possess anti-tumor, anti-ulcer, and hepatoprotective properties, as well as antibacterial and antiviral activities [2,3,4]. The biological activity of these preparations may result from the presence of vitamins, phenolic compounds, amino acids, unsaturated fatty acids, and micro- and macro-elements [5,6,7,8]. Moreover, the chemical composition of sea buckthorn is believed to be influenced by the choice of plant part, the variety or species, soil composition, and area of cultivation, among others [9].

The preparations from different organs of the sea buckthorn contain a range of chemical compounds, some of which may modulate hemostasis. For example, it has been observed that the phenolic fraction of sea buckthorn leaves is a rich source of hydrolysable tannins (mainly ellagitannins), while the non-polar fraction is a good source of triterpenes and various unidentified non-polar compounds. The phenolic fraction from the twigs of the studied plant is a source of proanthocyanidins and catechins, while the non-polar fraction is rich in triterpenes [10]. The chemical compositions of various sea buckthorn preparations are described in more detail in other reviews [11,12].

The present review paper describes the current literature concerning the effect of different preparations, i.e., extracts and fractions with various chemical compositions, as well as pure compounds obtained from various organs of the sea buckthorn, on hemostasis, and especially blood platelet functions; it also examines their positive role in the treatment and prevention of cardiovascular diseases. It sheds a new light on the mechanisms involved in their action on hemostasis both in vivo and in vitro. For these studies, biological materials, including blood platelets, plasma, and blood were isolated from healthy subjects and those with cardiovascular risk factors. In addition, it describes the cardioprotective potential of commercial products from different organs of sea buckthorn.

## 2. Hemostasis and Cardiovascular Diseases

Hemostasis is a complex sequence of reactions maintaining the fluidity of the blood circulating in the blood vessels and staunching the bleeding in the event of vessel disruption [13,14], with blood platelets, blood vessels, plasma coagulation factors, and the fibrinolytic system playing particularly important roles. Hemostasis can be divided into primary and secondary hemostasis. Primary hemostasis includes vascular hemostasis, i.e., the construction of a blood vessel due to damage to its wall and the secretion of serotonin; this is one mediator of many other processes, and this is followed by platelet hemostasis: platelet adhesion, reversible aggregation, release reaction (excretion of dense body content and alpha granules). The morphological changes taking place in the stimulated platelets are accompanied, among others, by the release of arachidonate and its transformation. Secondary hemostasis (plasma hemostasis) ends with the formation of fibrin [13,14].

Morbidity associated with cardiovascular disease has steadily grown in recent years and the condition is now classified as a civilization disease. The most common diseases of the circulatory system are coronary artery disease, stroke, arterial hypertension, atherosclerosis, and heart failure [15], with the main causative factors being high cholesterol, diabetes, obesity, smoking, and lack of physical activity. In addition, changes in elements of hemostasis are also observed. For example, smoking of tobacco products increases oxidative stress through the production of reactive oxygen species (ROS). Numerous studies indicate that ROS can modify individual components of the hemostatic system, including blood platelets, endothelial cells of blood vessels, and the plasma proteins of the coagulation and fibrinolysis systems [16,17].

Other studies show that an increased concentration of fibrinogen causes diseases of the cardiovascular system, and as such, reducing excessive amounts of fibrinogen may also reduce the mortality associated with cardiovascular diseases. Natural methods of lowering fibrinogen concentration include physical exercise, stopping smoking and alcohol consumption, weight reduction, and eating a proper diet [18,19]. In addition, quitting smoking is considered an effective method of reducing von Willebrand factor levels in people suffering from cardiovascular diseases [19,20]. An additional problem associated with cardiovascular disease is blood platelet hyper-reactivity. Patients suffering heart failure demonstrate higher blood platelet and coagulation system activity, with elevated fibrinogen and von Willebrand factor concentrations in the blood, and higher levels of P-selectin on the surface of activated vascular endothelial cells [16,21].

One of the anti-platelet drugs currently used in the therapy and prevention of cardiovascular diseases is acetylsalicylic acid, which inhibits the activity of cyclooxygenase. Aspirin is non-selective and irreversibly inhibits cyclooxygenase. It does so by acetylating the hydroxyl of serine residue. Cyclooxygenase is an important enzyme in the metabolism of arachidonic acid, in which is the formation of thromboxane A_2_–one of activators of platelet activation [22,23]. Unfortunately, various antiplatelet drugs, including acetylsalicylic acid, have various side effects, such as an increased risk of bleeding. Fruits, especially fresh berries, and their juices and extracts, may play an important role in the modulation of blood platelet activities [24], as well as preparations based on substances obtained from plants [22,25,26]. One group of compounds that significantly affects the circulatory system and oxidative stress are the phenolics, which occur in large amounts in some species, such as chokeberry fruits or grapes [27]. Research indicates that these compounds also inhibit the activity of cyclooxygenase, and that they prevent clotting by blocking the surface receptors for adhesion proteins (collagen, fibrinogen). Phenolic compounds also exhibit antioxidant properties: they have reducing properties, bind free radicals, and act as oxidase inhibitors and terminators that interrupt chain radical reactions. An additional advantage of supplementation with phenolic compounds is the lack of side effects [24,28,29,30].

## 3. Sea Buckthorn-Hemostasis and CVDs (In Vivo Trials)

Sea buckthorn can be consumed in various products; for example, fruits can be consumed in various forms, such as jams, juices, and wines. In studies on humans, dried fruits and preparations from sea buckthorn leaves are administered at doses of 500 to 2000 mg/day, while the oil is given at 2000 to 5000 mg/day as gelatin, liquids, and capsules (Sea buckthorn, available online, accessed on 11 October 2017). In vivo observations in rats, mice, and humans suggest that preparations and food products from sea buckthorn may reduce blood platelet activation and bestow cardiovascular benefits [31,32,33,34,35,36,37]. For example, sea buckthorn fruits and their food products, including juice, wine, and oil, have been found to have cardioprotective properties in humans [4,24,38,39,40,41].

In a recent study by Zhou et al. of the effect of sea buckthorn puree on cardiovascular disease risk factors in patients with hypercholesterolemia [42], patients (*n* = 56) consumed 30 g of sea buckthorn puree, made from frozen deseeded fruits, three times a day after meals. The results suggest that long-term administration lowered blood pressure and reduced inflammation parameters (for example C-reactive protein); however, the authors did not observe any changes in the concentrations of lipid parameters. In another experiment, Zhuo et al. [43] studied the effect of sea buckthorn flavone extracted from powdered leaves and whole fruits (75 mg/kg/day for 6 or 12 weeks) on atherosclerotic risk factors in mice. This preparation was administered orally by gastric canula. Although the authors did not identify the compounds in tested preparation, they only report that this preparation is flavone extract, and it inhibits macrophage foaming, inflammation, and vascular plaque formation via upregulation of adipokine C1q/tumor necrosis factor-related protein 6. More details about the effect of consumption of sea buckthorn products on cardiovascular diseases is given in Table 1. It is important to note that no adverse effects were observed in subjects treated with food products from sea buckthorn, including the oil [5].

## 4. Sea Buckthorn and Hemostasis—In Vitro Trials

### 4.1. Antioxidant Potential

Only a few human-based experiments appear to have examined the effects of sea buckthorn preparations on the oxidative stress associated with modulation of hemostasis and cardiovascular diseases. Blood platelets and plasma are often studied in vitro because they are important components of hemostasis. The influence of sea buckthorn preparations on the level of oxidative stress has been studied by Skalski et al., who examined lipid peroxidation product levels by thiobarbituric acid assay (TBA), as well as protein carbonyl and thiol group concentrations [44]. In these studies, oxidative stress was typically induced by hydrogen peroxide (H_2_O_2_) or the H_2_O_2_/Fe reaction mixture. The results found that the sea buckthorn leaf and twig extracts inhibit the peroxidation of plasma lipids caused by 15-min incubation with H_2_O_2_. When applied at doses of 5, 10, and 50 µg/mL for 15- and 60-min incubation, the twig extracts also decreased the level of H_2_O_2_/Fe induced plasma lipid peroxidation, with the activity of the extract being concentration dependent. The leaf extract also showed antioxidant properties at the same concentrations, but only with a 60 min incubation. Moreover, in this in vitro research model, the twig extract (50 µg/mL, incubation time-60 min) appeared to have stronger antioxidant effects than the leaf extract (50 µg/mL, incubation time-60 min) [44]. It was also found that the extracts from sea buckthorn had no effect on plasma protein carbonylation during the short incubation period (15 min); however, the leaf and twig extracts significantly inhibited H_2_O_2_/Fe-induced protein carbonylation during the longer incubation time (60 min). In addition, the twig extract (50 µg/mL, 60 min) had a stronger inhibitory effect on the carbonylation of human plasma proteins than the leaf extract (50 µg/mL, 60 min) [44]. Also, the tested leaf and twig extracts significantly reduced the level of oxidative stress in human plasma in vitro, with the stronger antioxidant activity demonstrated by the twig extract. It is likely that these differences were influenced by differences in the chemical profile of the examined extracts. For example, the twig extract has a high concentration of proanthocyanidins, which may be responsible for the strong antioxidant activity [44]. Other studies have shown that the saponin fraction from sea buckthorn leaves reduces oxidative stress in human plasma treated with H_2_O_2_/Fe (in in vitro trials) [45].

Studies on the effect of phenolic and non-polar leaf and twig fractions of sea buckthorn on oxidative stress parameters indicate that only the phenolic fraction isolated from the twigs significantly reduced the plasma lipid peroxidation stimulated by H_2_O_2_/Fe. Moreover, the non-polar fraction from the twigs (at the highest concentration tested-50 μg/mL) inhibited plasma lipid peroxidation. In this research model, the fraction rich in non-polar compounds isolated from the leaves also inhibited the lipid peroxidation process at all tested concentrations (0.5–50 μg/mL) [10]. It was also found that the phenolic fraction and the non-polar fraction isolated from sea buckthorn twigs appeared to protect against carbonylation of human plasma proteins induced by H_2_O_2_/Fe. Additionally, it was observed that the phenolic fractions from the leaves and twigs protect against oxidation of thiol groups in plasma proteins [10]. Hence, it was proposed that the high antioxidant potential of the phenolic fraction isolated from sea buckthorn twigs is correlated with the presence of proanthocyanidins and catechins. Proanthocyanidins are believed to be the most powerful natural antioxidant [10].

Olas et al. [46] report that sea buckthorn fruits are also natural sources of antioxidants. For example, the phenolic fraction, with a total concentration of phenolic compounds of 243 mg/g of fraction, inhibited plasma lipid peroxidation and protein carbonylation stimulated by H_2_O_2_ or H_2_O_2_/Fe. The major phenolic compounds of this fraction were flavonols, with two derivatives of isorhamnetin being present in the highest concentrations: isorhamnetin 3-O-beta-glucoside-7-O-alpha-rhamnoside and isorhamnetin 3-O-beta-glucoside-7-O-alpha-(3″-isovaleryl)–rhamnoside. The authors propose that these flavonols may act as the main antioxidants in the phenolic fraction from sea buckthorn fruits. In addition, it was noted that 30 min of incubation of plasma with isorhamnetin and its two derivatives isolated from sea buckthorn fruits reduces H_2_O_2_/Fe-stimulated lipid peroxidation. Interestingly, the phenolic fraction of the fruits had no significant influence on this process.

Similar results were obtained in studies determining carbonyl group level in human plasma: in this system, isorhamnetin and its two derivatives significantly inhibited protein carbonylation. In contrast, while the two tested isorhamnetin derivatives and the phenolic fraction were found to have protective properties based on the thiol group level, isorhamnetin itself had no such effect. In this case, it is assumed that the flavonoids (isorhamnetin and its two derivatives) act as free radical scavengers [47].

It is known that ROS can act as secondary signaling molecules. They are generated in unstimulated and thrombin-stimulated blood platelets [48]. For example, the formation of the superoxide radical anion (O_2_^−^) is correlated with the enzymatic pathway of arachidonic acid metabolism. The effect of sea buckthorn preparations on the O_2_^−^ level was determined using a method based on the reduction of cytochrome c in washed blood platelets. It was found that only the twig extract significantly reduced the production of superoxide anion in unstimulated and thrombin-stimulated blood platelets [49].

An identical test system was used for evaluating the phenolic and non-polar fractions from sea buckthorn leaves and twigs. All the fractions used (1 and 10 µg/mL) significantly reduced the amount of O_2_^−^ in unstimulated and thrombin-activated platelets [50]. It can be concluded that all of them are sources of compounds that can modulate the activity of platelets by interfering with an arachidonic acid metabolism [49,50]. Changes in platelet responsiveness to thrombin were also observed when platelets were preincubated with the phenolic fraction from sea buckthorn fruits. It is possible that this fraction modulates the production of ROS by interfering with the arachidonic acid metabolism, in which cyclooxygenase takes part. The used fraction was tested at a concentration range of 0.5–50 µg/mL, which corresponds to the physiological range of phenolic compounds in human plasma [51].

It is an important that phenolic compounds, including flavonoids, can act also as prooxidants in cells [52]. Their prooxidative effect may be relevant in vivo if free transition metal ions are involved in the oxidation processes. Flavonoids are capable of reducing Cu(II) to Cu(I) and thus enable the formation of free radicals. It is known that tissue damage can cause the release of iron or copper. In addition, the presence of metal ions has been observed in atherosclerotic lesions [53]. In such cases, the possibility of flavonoids acting as prooxidants cannot be overlooked [54]. Bakir et al. [55] have observed prooxidant effects of isoramnetin on linoleic acid peroxidation stimulated by Cu(II) and H_2_O_2_. Moreover, flavonoids can also nick DNA via the production of radicals in the presence of Cu and O_2_ [52].

### 4.2. Sea Buckthorn Preparations and Plasma Hemostasis

Sea buckthorn preparations may have an effect on plasma hemostasis in vitro [56]. Blood clot formation was treated with different sea buckthorn fractions in a real-time hydrodynamic blood flow model and studied using Total Thrombus-formation Analysis System (T-TAS), used to assess thrombogenicity in whole blood. In this experiment, a collagen-coated chip was used to visualize the formation of a platelet thrombus. Whole blood was incubated with the tested preparations, i.e., phenolic and nonpolar compounds from sea buckthorn fruits, leaves, and twigs (37 °C; 30 min). It was noticed that the phenolic fraction of the fruits, leaves, and twigs of the tested plant significantly slowed down the process of thrombus formation, and that the fraction rich in non-polar compounds from the leaves inhibited this process [56]. Recently, Juszczak et al. [57] reported that the saponin fraction from sea buckthorn leaves appears to demonstrate anticoagulant potential at a concentration of 50 µg/mL, as measured in human whole blood using T-TAS. This property may be helpful in reducing prothombotic states.

The effect of various sea buckthorn preparations on blood clotting times can also be measured based on thrombin time (TT), prothrombin time (PT), and activated thromboplastin time (APTT). The findings provide an insight into whether the tested preparations influence plasma hemostasis, and at what stage, and whether the tested preparations prolong or shorten blood clot formation. Skalski et al. [44] report that sea buckthorn twig and leaf extracts (at the highest tested concentration-50 μg/mL) incubated with human plasma for 30 min at 37 °C significantly extended APTT time: APTT being a measure of the effectiveness of the mechanism of coagulation without the involvement of platelets. The twig extract was also shown to have a stronger anticoagulant effect than the leaf extract. Additionally, the twig extract had a stronger anticoagulant effect than the sea buckthorn extract. The tested preparations did not significantly change TT and PT times [44].

Moreover, an analysis of the influence of phenolic and non-polar fractions from sea buckthorn leaves and twigs (in the dose range 0.5–50 μg/mL; incubation time 30 min) on plasma coagulation properties showed that the phenolic fraction from leaves significantly extended PT time. It is presumed that the anticoagulant properties of this fraction may be related to its modulation of the activity of prothrombin or coagulation factors V, VII, and X. In addition, the nonpolar fraction of the twigs also significantly increased the activated thromboplastin time. This activity is probably related to the presence of triterpenes, acylated triterpenes, and unidentified polar compounds. None of the studied fractions changed the TT time [10].

An analysis of the effect of isorhamnetin, its two derivatives (isolated from the phenol fraction of sea buckthorn berries) and the fraction rich in phenolic compounds from sea buckthorn fruits showed that only isorhamnetin 3-O-beta-glucoside-7-O-alpha-(3″-isovaleryl)-rhamnoside significantly extended TT. This activity is presumed to be independent of the modulation of thrombin activity. In contrast, Choi et al. [58] noted that flavonoids can inhibit the enzymatic activity of thrombin. Thrombin, as a serine protease, not only plays an important role in coagulation processes, but is also an activator of blood platelets [58]. The tested fractions did not appear to have any significant effect on other clotting times [47].

### 4.3. Sea Buckthorn Preparations and Blood Platelets

The influence of sea buckthorn preparations on selected markers of blood platelet activation has been analyzed using different biological materials: isolated washed blood platelets, platelet-rich plasma (PRP) and whole blood (Table 2) [49,50,56]. Changes in the activation of blood platelets in the presence of these preparations were analyzed by (1) colorimetric measurement of the adhesion of blood platelets to collagen and fibrinogen, (2) turbidimetric measurement of platelet aggregation, (3) flow cytometry measurement of P-selectin and GPIIb/IIIa receptor exposition on the surface of platelets in whole blood, and (4) measurement of VASP phosphorylation (vasodilator-stimulated phosphoprotein).

Skalski et al. [49] report significantly lower adhesion of unstimulated and thrombin-stimulated platelets to collagen type I, i.e., the most prominent collagen type in the arterial wall in vessels changed by atherosclerosis, in the presence of extracts from leaves and sea buckthorn twigs (0.5–50 μg/mL). At the highest concentration, the sea buckthorn twig extract caused a greater inhibition of adhesion of thrombin-activated platelets to collagen and fibrinogen than the leaf extract. Moreover, pre-incubation with the phenolic fractions from twigs (1 µg/mL) and leaves (1; 10 µg/mL), and the non-polar fractions from twigs (1; 10 µg/mL) and leaves resulted in significant inhibition of the adhesion of unstimulated platelets to collagen at all tested concentrations (1–50 μg/mL). The phenolic fraction from the twigs inhibited adhesion of thrombin-stimulated platelets to collagen at all tested concentrations (1, 10 and 50 μg/mL), while the phenolic fraction from the leaves significantly inhibited the adhesion at concentrations of 1 and 50 μg/mL. In addition, it was observed that preincubation with all tested fractions, viz. the phenolic and the non-polar fractions from twigs and leaves, at all tested concentrations (1–50 μg/mL), significantly inhibited the adhesion of platelets to fibrinogen following thrombin and ADP stimulation [50]. Olas et al. [51] also indicate that the phenolic fraction (0.5–50 μg/mL) from sea buckthorn fruits appears to have anti-adhesive properties, but these were weaker when ADP was used as the agonist than when thrombin was used. The authors propose that the extract reduces the level of ROS, which may act as secondary messengers, in platelets activated by thrombin.

Other studies examined the anti-aggregation potential of the preparations (extracts and fractions) from leaves and sea buckthorn twigs [49,50]. Blood platelet aggregation was induced by various agonists: ADP, collagen, and thrombin. Although the tested extracts were not found to show any anti-aggregation properties when ADP and collagen were used, the leaf extract (10 and 50 μg/mL) significantly inhibited the aggregation of thrombin-treated platelets, as did the twig extract at the highest tested concentration [49]. Skalski et al. also report that ADP-stimulated platelet aggregation was significantly inhibited by the phenolic fractions from twigs and the leaves, and the non-polar fractions from the twigs and the leaves. However, none of the tested fractions appeared to have any significant effect on the aggregation of platelets treated with collagen or thrombin. It was also noted that only the phenolic fraction from the leaves, at the highest concentration (50 μg/mL), significantly inhibited ADP-induced platelet aggregation [50]. The authors therefore propose that the tested fractions isolated from sea buckthorn leaves and twigs have anti-platelet properties.

In addition, various studies confirm that proanthocyanidins and anthocyanidins have a positive effect on the cardiovascular system by inhibiting the activation of platelets [24,60]; as such, these compounds may determine the anti-platelet properties of the phenolic fractions from sea buckthorn. However, the mechanisms of the antiplatelet action of triterpenoids and their derivatives presented in non-polar fractions are not fully understood and require further research.

However, Skalski et al. [47] report that isorhamnetin and its derivative-isorhamnetin 3-O-beta-glucoside-7-O-alpha- (3″-isovaleryl)-rhamnoside (isolated from sea buckthorn fruits) appear to inhibit thrombin-stimulated platelet aggregation. Moreover, these compounds seem to have no effect on the aggregation of platelets following ADP and collagen stimulation; suggesting that may modulate platelet activation by interfering with the action of the thrombin receptors on the platelets [47]. Cheng et al. [59] note that flavones isolated from sea buckthorn fruits (3 μg/mL) demonstrate anti-aggregatory potential in vitro, but only when collagen was used as the platelet activator: no such anti-aggregatory action was observed in the presence of ADP and arachidonic acid.

Platelet activation is associated with the formation of thromboxane A_2_ in the arachidonic acid pathway. In one study based on TBARS concentration, an indicator of enzymatic peroxidation of arachidonic acid in platelets, the sea buckthorn twig extract was found to significantly reduce lipid peroxidation in thrombin-treated platelets. It is therefore proposed that phenolic compounds present in the extract inhibit the activation of the arachidonic acid pathway [49].

In other studies, the phenolic and non-polar fractions from leaves and twigs were found to significantly inhibit enzymatic peroxidation of lipids in platelets following thrombin stimulation at all tested concentrations (1 to 50 µg/mL) [50]. This may suggest that the bioactive compounds present in the tested fractions are able to modulate the activity of platelets by interfering with arachidonic acid metabolism and may affect platelet reactivity by modifying the level of ROS and modulating the exposition of platelet receptors [50]. Three-color flow cytometry showed that samples treated with various fractions of sea buckthorn extract demonstrated altered levels of platelet receptors. For example, the phenolic and nonpolar fruit and leaf fractions decreased in 10 µM ADP-activated platelets, while the nonpolar twigs fraction increased PAC-1 binding in 20 µM ADP-activated platelets. The phenolic fraction (5 and 50 µg/mL) from fruit was found to reduce PAC-1 binding in collagen-activated platelets. It can be assumed that the inhibition of platelet aggregation, confirmed in previous studies [49,50], is correlated with low activation of the GPIIb/IIIa receptor [56].

Two purinergic receptors, viz. P_2_Y_1_ and P_2_Y_12_, are known to be involved in the activation of ADP-stimulated platelets. Phosphorylation of VASP correlates with inhibition of the P_2_Y_12_ receptor, while the state of no phosphorylation correlates with its activation. The interaction between antiplatelet drugs (for example, clopidogrel) with the P_2_Y_12_ receptor is studied using VASP phosphorylation monitoring assay. Skalski et al. [56] monitored the platelet-specific ADP receptor (P_2_Y_12_) with a flow cytometry kit (PLT VASP/P_2_Y_12_). The results are reported as PRI (Platelet Reactivity Index). No differences were observed between the PRI values of the samples treated with the tested sea buckthorn fractions isolated from fruits, leaves, and twigs (50 µg/mL) and the control samples, which may indicate that the antiplatelet potential of the tested fractions is not dependent on the P_2_Y_12_ receptor [56].

Based on the findings [47,49,50,56,59], it is important to note that the effect of sea buckthorn preparations on blood platelets appears to be dependent on a range of inductor of platelets (for example, thrombin, collagen, and ADP), the type of preparation, its chemical content (for example class of phenolic compounds), and its concentration. The inhibitory action of sea buckthorn preparations can be influenced also by the amount of platelet agonist used to induce platelet activation.

It is important to note that none of the tested preparations were toxic to platelets (measured as extracellular lactate dehydrogenase activity) [49,50]. Indeed, other studies have also demonstrated that preparations from various parts of sea buckthorn are safe; for example, the hydro-alcoholic extract of sea buckthorn leaf extract was found to be non-toxic in rats and mice up to doses of 15 g/kg of body weight [61], and the aqueous extract of sea buckthorn fruits (100 mg/kg body weight/day for 90 days) was also not toxic in rats [62]. While the Chinese Pharmacopeia indicates that a safe dose for use of sea buckthorn dried fruits is approximately 3–10 g/day for humans [63], little further information exists regarding the safety of sea buckthorn for humans; however, the present findings may suggest that sea buckthorn and its products are safe for consumption by humans in food and in dietary supplements [64].

The mechanism by which the preparations from sea buckthorn exert their inhibitory activity on blood platelet activation is complex and still unknown; however, it is possible that these preparations induce a decrease in ROS production, and the resulting fall in ROS level may be accompanied by a recovery of phosphatase activity (Figure 1). The reactivated phosphatases may inhibit tyrosine kinase activity and blood platelet aggregation by inhibition of GPIIb/IIIa exposition. Sea buckthorn preparations also reduce the activity of COX and inhibit arachidonic acid metabolism; however, the antiplatelet potential of the preparations is not dependent on the P2Y12 receptor.

## 5. Medicines and Food Products from Sea Buckthorn

Due to its unique chemical composition, sea buckthorn has become a common ingredient in connection with cosmetics and food products. In addition, due to its known health-promoting properties, numerous dietary supplements are also available on the market. Singh et al. [61] and Wang et al. [64] report that hundreds of commercial products containing sea buckthorn are available in various countries, including China, India, the United States, and European countries, such as Germany and Finland. However, the most consumed part of the plant are the fruits. They are classified as having a fatty nature, i.e., the oil from peel, seeds, and pulp, or an aqueous nature, for example, clarified juice. Regarding food, sea buckthorn fruits have grown from being additives to being foods in their own right; it is now possible to find 100% sea buckthorn jams [65]. Other food products, such as yoghurt, cheese, and beverages, already benefit from the addition of sea buckthorn fruits. Sea buckthorn teas are also available as 50:50 mixtures of fruit and leaves and as the leaves alone, which have been proposed as a great supplement for losing weight [1,65]. Recently, Ciesarova et al. [9] suggested that consumer acceptance of sea buckthorn food products mainly depends on the sugar/acid ratio. Wang et al. [65] compiled a database of chemical compounds in sea buckthorn, which contains 74 bioactive compounds and 106 nutrients; they also note 17 commonly-marketed sea buckthorn products from eight countries, including sea buckthorn juice, sea buckthorn leaf tea, sea buckthorn powder, sea buckthorn oil, and sea buckthorn capsules.

There is currently great interest in creating formulas based on sea buckthorn that may benefit cardiovascular status. A review by Singh et al. [61] cites six such patents (Table 3). The first Chinese patent [66] is a formula rich in flavonoids, hydroxypropyl, and linoleic acid. In tests on a group of 160 patients suffering from coronary heart disease (50–76 years) over a three-month period, the formula was found to have better treatment effectiveness than isorbide mononitrate and Xindakang (SBT flavone-based formulation). Another Chinese patent [67], which arose as an extension of the research above, is a rich source of isorhamnetin, quercetin, and kaempferol. Administration to a group of 200 outpatients, divided into four study groups, for three months found it to be effective in the treatment of angina pectoris. The third Chinese patent [68] concerns a formula rich in fatty acids and phytosterol. Studies on a group 160 hyperlipidemic patients found it to have better hypolipidemic effects than controls, with total cholesterol, triglyceride, high density lipoprotein, and low density lipoprotein (LDL) being assessed. Another Chinese patent [69] containing formulas rich in tryptamine derivatives (sea buckthorn seeds) was found to have anti-ischemic activity [61]. Finally, a Chinese patent [70] concerning a formula (see buckthorn leaves) rich in flavonoids. Fourteen-day administration to Sprague Dawley rats resulted in a significant reduction of total cholesterol, LDL, and triglyceride content and improved blood lipids [61]. In addition, a recent European patent [68] described a formula rich in β-D-glucopyranose, L-quebrachitol, phenolic acids, proanthocyanidins and peptides: this was found to inhibit microsomal lipid peroxidation and LDL oxidation in vitro [61]. The variety of sea buckthorn food products and their cardiovascular potential is summarized in Figure 2.

## 6. Conclusions

The studies described in this review paper examine the effect of different sea buckthorn preparations, and the pure compounds extracted from them, on hemostasis both in vivo and in vitro. For these studies, biological materials, including blood platelets, plasma, and blood were isolated from healthy subjects and those with cardiovascular risk factors. Based on the findings, it is important to note that the effect of sea buckthorn preparations on hemostasis appears to be dependent on a range of factors, including the type of preparation, its chemical content, and its concentration. Owing to their high phytochemical contents, especially phenolic compounds, the sea buckthorn preparations appear to be effective regulators of hemostasis, particularly blood platelet function. However, many studies present the chemical composition of sea buckthorn preparations as their share of the total area of CAD peaks (Charged Aerosol Detector) rather than as concentration per se (expressed e.g., in mg/g), and are hence, only indicative.

The effects of sea buckthorn preparations on hemostasis and CVDs in different in vitro and in vivo models are summarized in Table 1 and Table 2, but the number of these models is too limited to unequivocally indicate that the sea buckthorn preparations have beneficial effects on hemostasis and cardiovascular diseases. In addition, the effect of sea buckthorn preparations, especially commercial products, on human hemostasis and CVDs have not been well described in scientific literature.

Different studies have employed various types of preparations, the composition of which was not always determined, and their bioactive ingredients were, in many cases, not clearly identified.

In addition, the prophylactic and treatment doses of these preparations are currently unknown, and the recommendations for the use of sea buckthorn preparations are very often based on small clinical trials. Nevertheless, these preparations may represent a promising alternative to classical drugs or supplements with antiplatelet activity and some patents exploiting the cardiovascular activity of sea buckthorn are demonstrated in Table 3. Therefore, more randomized clinical trials with larger groups are needed, especially with healthy people, and people with the highest cardiovascular risk factors. These trials should also determine the influence of various sea buckthorn compounds on hemostasis, including not only blood platelet functions, but also coagulation systems and fibrinolysis. It is also important to explore the role of different sea buckthorn products in the prophylaxis and treatment of cardiovascular diseases, because no credible evidence for anti-hemorrhagic efficiency of sea buckthorn preparations in humans or animals have been reported.

## Figures and Tables

**Figure 1 nutrients-14-00991-f001:**
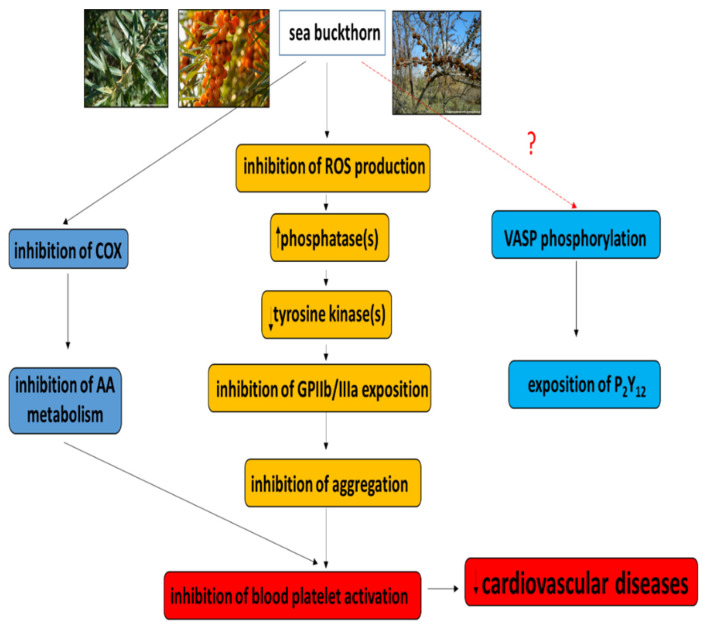
Proposed mechanism of action of preparations from sea buckthorn on blood platelets. These preparations induce a decrease in reactive oxygen species (ROS) production. The decrease in intracellular ROS level may be accompanied by the recovery of phosphatase activity. Reactivated phosphatases may inhibit the activity of tyrosine kinases, and thus platelet aggregation (by inhibition of GPIIb/IIIa exposition). Sea buckthorn preparations also reduce the activity of cyclooxygenase (COX) and reduce arachidonic acid (AA) metabolism. However, the antiplatelet potential of the preparations is not dependent on the P_2_Y_12_ receptor.

**Figure 2 nutrients-14-00991-f002:**
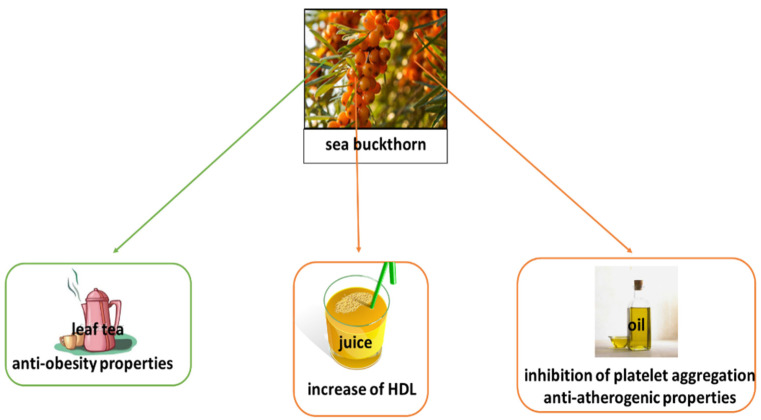
Sea buckthorn food products and their cardiovascular potential.

**Table 1 nutrients-14-00991-t001:** The effect of consumption of sea buckthorn products on parameters of cardiovascular diseases in in vivo studies.

Sea Buckthorn Product	Dose/Days	Subjects	Activity of Sea Buckthorn Product	References
Juice	300 mL/day (56 days)	30 Healthy people (male non-smokers, aged 18–55 years)	Increase in HDL concentration	[38]
Pulp and seed oil	Ten 500 mg capsules/day (42 days)	Healthy people (aged 20–59 years, body mass index 19.6–26.5 kg/m^2^)	Inhibition of blood platelet aggregation	[31]
Fruit puree	30 g/day (90 days)	Healthy people (males aged 50–70 years old or postmenopausal females)	Increase in HDL concentration	[42]
Seed oil	5 mL/day (60 days)	Adult New Zealand white rabbits (2.50–1.0 kg bodyweight)	Decrease in total cholesterol	[32]
Dry fruits (powder)	0.7 g/kg/day (60 days)	Stroke-prone rats	Decrease in total plasma cholesterol, triglyceride, heart rate, and blood pressure	[34]
Pulp oil	5, 10 and 20 mL/kg/day (30 days)	Wistar rats (males, albino)	Protecting against myocardial ischemia-reperfusion injury	[37]
Total flavones extracted from seed residues	150 mg/kg/day (42 days)	Chronic sucrose-fed rats	Antihypertensive action	[33]
Tea from leaves	1 or 5%/day (42 days)	Obese mice	Anti-visceral obesity potential	[35]
Ethanolic extract of leaves	500 or 1000 mg/kg	C57BL/6J mice (male)	Anti-obesity potential	[36]
Flavone (obtained from powdered leaves and whole fruits)	75 mg/kg/day (42 or 84 days)	The apoE deficient mouse	Inhibition of macrophage foaming, inflammation, and vascular plaque formation	[43]

**Table 2 nutrients-14-00991-t002:** The effect of preparations from various organs of the sea buckthorn on selected elements of hemostasis (in vitro).

Element of Hemostasis	Sea Buckthorn			Reference
	Fruits	Leaves	Twigs	
Inhibition of blood platelet adhesion to collagen and fibrinogen (using washed human blood platelets)	+	+	+	[50,51]
Inhibition of blood platelet aggregation stimulated by thrombin (using washed human blood platelets)	?	-	-	[50,51]
Inhibition of blood platelet aggregation stimulated by ADP (using human platelet rich-plasma)	-	-	+	[50,51]
Inhibition of blood platelet aggregation stimulated by collagen (using human platelet rich-plasma)	?	-	-	[50,51,59]
Inhibition of eicosanoid synthesis (using human washed blood platelets)	+	+	+	[10,46]
Antioxidant activity (using human washed blood platelets and plasma)	+	+	+	[10,45,46]
Reduction of GPIIb/IIIa exposition (using human whole blood)	+	+	+	[56]
Reduction of thrombus formation (using human whole blood)	+	+	+	[56,57]

“?”—no studied; “+”—effect existing; “-”—no effect.

**Table 3 nutrients-14-00991-t003:** Patents on cardiovascular activity of sea buckthorn (based on Singh et al. [61]).

Number and Name of Patent	Chemical Content	Dose/Days	Subjects
A Chinese patent (CN103505484A)	Flavonoids, hydroxypropyl, and linoleic acid	20 mg daily	Patients with coronary heart disease; angina pectoris
A Chinese patent (CN103505451A)	Isorhamnetin, quercetin, and kaempferol	20 mg daily	Patients with coronary heart disease; angina pectoris
A Chinese patent (CN103505483A)	Fatty acids, phytosterol	-	Patients with hyperlipidemic
A Chinese patent (CN101612176A)	Tryptamine derivatives	-	Ischemic myocardial cells, in vitro
A Chinese patent (CN102058631A)	Flavonoids	1, 2, 4 g/kg bw for 14 days	Sprague Dawley rat model

## Data Availability

Not applicable.

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
