# Peer review of "Preparations from Various Organs of Sea Buckthorn (Elaeagnus rhamnoides (L.) A. Nelson) as Important Regulators of Hemostasis and Their Role in the Treatment and Prevention of Cardiovascular Diseases"

_nutrients, 2022, doi:10.3390/nu14050991_

Round 1

Reviewer 1 Report

Title should be revised to be more exact - hemostasis is only a partial topic, a substantial part also deals with cardiovascular diseases.

Abstract: Instead of the first part (what is too general description of the sea buckthorn usage), the authors should state the main contribution of their review. Especially, the last sentence of the abstract should be a strong summary statement of the review.

At the end of the introduction section (as well as of abstract), there is statement that “It also sheds new light on the mechanisms involved in their action” - so, this needs to be emphasized more…

Table 1 is not necessary in the review that is focused also (maybe mainly) on leaf and twig extracts. Or table summarizing the bioactive compounds in all organs discussed in this review could be more relevant and useful to readers. The proposed table can be linked to the chapter “Antioxidant potential” (lines 164 - 225). This chapter is to high extent not focused on hemostasis - I suggest the authors to transfer its shortened version in the introduction section.

Some additional parts are also too general and may be shortened - e.g. introduction, chapter “Hemostasis and cardiovascular diseases” …

In my opinion, the review has three main parts (or focuses):

  1. general characteristics + antioxidant action of sea buckthorn organs
  2. sea buckthorn effect in cardiovascular diseases
  3. sea buckthorn effect on hemostasis.

To increase clarity and comprehensibility of the review, each of these orientations can be addressed consecutively and separately.

Is the link to Tab.2 - line 294 - correct?

Unify writing: table 2, tab. 1 - as well as in the text.

Not all abbreviations are defined (e.g. T-TAS). If you define abbreviations, you are supposed to use it - e.g. pro/thrombin time.

There are notes in the text that none of the sea buckthorn preparations were toxic. How was the potential toxicity tested?

In table 2 - the heading of the fourth column - “Parameters of cardiovascular diseases” in not accurate - it deals with the effects of mentioned products. What exactly is meant by the “parameters of cardiovascular diseases”? Increase of HDL?

In table 4, column “Subjects and parameters of cardiovascular diseases” - in some cases there is description of effects, in some of them not. Please unify.

In figure 1, the arrow leading to VASP phosphorylation may be misleading. Or indicate another way that there is not known relation (question mark is not sufficient).

In figure 2 - in the box with “lyophilized powder” - there is no mention regarding its cardiovascular potential.

The author’s personal view on this topic as well as weak points (e.g. when administration of sea buckthorn extracts are not accompanied by data about alterations in observed parameters) are not summarized in this article.

Based on available data included in this review it would be interesting to know what exactly the authors consider as a challenging to investigate for further progress in this field.

Chapter “Conclusions” mostly recapitulates what has been written several times before. Instead, short summary or 1-2 sentences with the most important message is recommended.

Author Response

We would like to thank the Reviewer for providing helpful comments.

Title should be revised to be more exact - hemostasis is only a partial topic, a substantial part also deals with cardiovascular diseases.

Response: We have changed the title. Now, it is: “Preparations from various organs of sea buckthorn (Elaeagnus rhamnoides (L.) A. Nelson) as important regulators of hemostasis and their role in the treatment and prevention of cardiovascular diseases” 

Abstract: Instead of the first part (what is too general description of the sea buckthorn usage), the authors should state the main contribution of their review. Especially, the last sentence of the abstract should be a strong summary statement of the review.

Response: We have changed the abstract.

At the end of the introduction section (as well as of abstract), there is statement that “It also sheds new light on the mechanisms involved in their action” - so, this needs to be emphasized more…

Response: We have added more information: “It sheds a new light on the mechanisms involved in their action on hemostasis both in vivo and in vitro. For these studies, biological materials, including blood platelets, plasma and blood were isolated from healthy subjects and those with cardiovascular risk factors. In addition, it describes the cardioprotective potential of commercial products from different organs of sea buckthorn.”

Table 1 is not necessary in the review that is focused also (maybe mainly) on leaf and twig extracts. Or table summarizing the bioactive compounds in all organs discussed in this review could be more relevant and useful to readers. The proposed table can be linked to the chapter “Antioxidant potential” (lines 164 - 225). This chapter is to high extent not focused on hemostasis - I suggest the authors to transfer its shortened version in the introduction section.

Response: We have not removed Table 1, because sea buckthorn can be consumed in various products; for example, the fruits can be consumed in various forms, such as jams, juices and wines. For example, sea buckthorn fruits and their food products, including juice, wine and oil, have been found to have cardioprotective properties in humans (4,24,38-41). The effect of consumption of sea buckthorn products on cardiovascular diseases in in vivo models is given in Table 1. 

It has known that the oxidative stress may modulate hemostasis. Therefore, we have not transfer the chapter of “Antioxidant potential” to the Introduction.             

Some additional parts are also too general and may be shortened - e.g. introduction, chapter “Hemostasis and cardiovascular diseases” …

In my opinion, the review has three main parts (or focuses):

  1. general characteristics + antioxidant action of sea buckthorn organs
  2. sea buckthorn effect in cardiovascular diseases
  3. sea buckthorn effect on hemostasis.

To increase clarity and comprehensibility of the review, each of these orientations can be addressed consecutively and separately.

Response: We hope that our manuscript is clarity. In our manuscript, there are different chapters:

  • Introduction: general characteristics of sea buckthorn
  • Hemostasis and CVDs: general characteristics
  • Sea buckthorn - hemostasis and CVDs (in vivo trials)
  • Sea buckthorn and hemostasis – in vitro trials
  • Medicines and food products from sea buckthorn
  • Conclusion

Is the link to Tab.2 - line 294 - correct?

Response: We have corrected the link to Tab. 2.

Unify writing: table 2, tab. 1 - as well as in the text.

Response: We have corrected.

Not all abbreviations are defined (e.g. T-TAS). If you define abbreviations, you are supposed to use it - e.g. pro/thrombin time.

Response: We have corrected, for example, T-TAS – Total Thrombus-formation Analysis System. We have also used abbreviations: PT and TT in text of manuscript.

There are notes in the text that none of the sea buckthorn preparations were toxic. How was the potential toxicity tested?

Response: We have added more information about it: “It is important to note that none of the tested preparations were toxic to platelets (measured as extracellular lactate dehydrogenase activity).

In table 2 - the heading of the fourth column - “Parameters of cardiovascular diseases” in not accurate - it deals with the effects of mentioned products. What exactly is meant by the “parameters of cardiovascular diseases”? Increase of HDL?

Response: We have corrected this table.

In table 4, column “Subjects and parameters of cardiovascular diseases” - in some cases there is description of effects, in some of them not. Please unify.

Response: We have corrected this table.

In figure 1, the arrow leading to VASP phosphorylation may be misleading. Or indicate another way that there is not known relation (question mark is not sufficient).

Response: We have corrected this figure.

In figure 2 - in the box with “lyophilized powder” - there is no mention regarding its cardiovascular potential.

Response: We have corrected this figure.

The author’s personal view on this topic as well as weak points (e.g. when administration of sea buckthorn extracts are not accompanied by data about alterations in observed parameters) are not summarized in this article.

Based on available data included in this review it would be interesting to know what exactly the authors consider as a challenging to investigate for further progress in this field.

Chapter “Conclusions” mostly recapitulates what has been written several times before. Instead, short summary or 1-2 sentences with the most important message is recommended.

Response: Details about the effect of consumption of sea buckthorn products on cardiovascular diseases is given in Table 1.  In addition, the influence of sea buckthorn preparations on selected markers of blood platelet activation has been analyzed using different biological materials: isolated washed blood platelets, platelet-rich plasma (PRP) and whole blood in in vitro models and was presented in Table 2. Moreover, the mechanism by which the preparations from sea buckthorn exert their inhibitor activity on platelet activation was presented on Figure 1. However, this mechanism is complex and still unknown; however, it is possible that these preparations induce a decrease of ROS production, and the resulting fall in ROS level may be accompanied by a recovery of phosphatase activity (Figure 1).

We have also changed the conclusion.

Reviewer 2 Report

Congratulations on the extensive documentation. It is necessary to underline the irreversible bond with the cyclooxygenases of acetyl salicylic acid and describe the parallel mechanism of action of the active principle described, which would act and if the bond would be irreversible page 3 line 127

Author Response

Congratulations on the extensive documentation. It is necessary to underline the irreversible bond with the cyclooxygenases of acetyl salicylic acid and describe the parallel mechanism of action of the active principle described, which would act and if the bond would be irreversible page 3 line 127

Response:

We would like to thank the Reviewer for providing helpful comments.

We have added more information about it: “Aspirin is non-selective and irreversibly inhibits cyclooxygenase. It does it by acetylating the hydroxyl of serine residue.”